# Quantitative Microbial Risk Assessment Applied to *Legionella* Contamination on Long-Distance Public Transport

**DOI:** 10.3390/ijerph19041960

**Published:** 2022-02-10

**Authors:** Ileana Federigi, Osvalda De Giglio, Giusy Diella, Francesco Triggiano, Francesca Apollonio, Marilena D’Ambrosio, Lorenzo Cioni, Marco Verani, Maria Teresa Montagna, Annalaura Carducci

**Affiliations:** 1Laboratory of Hygiene and Environmental Virology, Department of Biology, University of Pisa, Via S. Zeno 35/39, 56127 Pisa, Italy; marco.verani@unipi.it (M.V.); annalaura.carducci@unipi.it (A.C.); 2Department of Biomedical Science and Human Oncology, University of Bari Aldo Moro, Piazza G. Cesare 11, 70124 Bari, Italy; osvalda.degiglio@uniba.it (O.D.G.); giusy.diella@uniba.it (G.D.); francesco.triggiano@uniba.it (F.T.); francesca.apollonio@uniba.it (F.A.); marilena.dambrosio@uniba.it (M.D.); 3Scuola Normale Superiore, Piazza dei Cavalieri 7, 56126 Pisa, Italy; lorenzo.cioni@sns.it; 4Regional Reference Laboratory of Clinical and Environmental Surveillance of Legionellosis, Department of Biomedical Science and Human Oncology, University of Bari Aldo Moro, Piazza G. Cesare 11, 70124 Bari, Italy; mariateresa.montagna@uniba.it

**Keywords:** *Legionella pneumophila*, long-distance public transport, train, monitoring, bioaerosol, water distribution system, premise plumbing, train, risk assessment, public health

## Abstract

The quantitative microbial risk assessment (QMRA) framework is used for assessing health risk coming from pathogens in the environment. In this paper, we used QMRA to evaluate the infection risk of *L. pneumophila* attributable to sink usage in a toilet cabin on Italian long-distance public transportation (LDT). LDT has water distribution systems with risk points for *Legionella* proliferation, as well as premise plumbing for drinking water, but they are not considered for risk assessment. Monitoring data revealed that approximately 55% of water samples (217/398) were positive for *L. pneumophila*, and the most frequently isolated was *L. pneumophila* sg1 (64%, 139/217); therefore, such data were fitted to the best probability distribution function to be used as a stochastic variable in the QMRA model. Then, a sink-specific aerosolization ratio was applied to calculate the inhaled dose, also considering inhalation rate and exposure time, which were used as stochastic parameters based on literature data. At *L. pneumophila* sg1 concentration ≤100 CFU/L, health risk was approximately 1 infection per 1 million exposures, with an increase of up to 5 infections per 10,000 exposures when the concentrations were ≥10,000 CFU/L. Our QMRA results showed a low *Legionella* infection risk from faucets on LDT; however, it deserves consideration since LDT can be used by people highly susceptible for the development of a severe form of the disease, owing to their immunological status or other predisposing factors. Further investigations could also evaluate *Legionella*-laden aerosols from toilet flushing.

## 1. Introduction

Built environments for residential, tourist accommodation, healthcare, and long-distance public transportation (LDT) are equipped with various types of water storage and distribution systems for hygienic purposes and safe removal of human waste [1]. Apart from their design, the moist engineered surfaces of pipes and water tanks can be sensitive to biofilm growth; thus, they represent an ecological niche for environmental bacteria, which can behave as opportunistic pathogens, such as nontuberculous mycobacteria, *Pseudomonas* spp., and *Legionella* spp. [2,3]. Particular attention is dedicated to *Legionella* spp. since *Legionella pneumophila* is one of the main etiological agents of epidemic pneumonia associated with water systems throughout the world [4,5,6]. Many environmental studies isolated *L. pneumophila* in water from premise plumbing, which refers to all piping located downstream of the main water distribution system and within buildings, for example collecting water samples from hot/cold mixing valves in various settings, such as healthcare facilities [7,8], hotels [9,10], cruise ships [11,12], retirement homes [13], private homes, or office buildings [14,15,16]. Moreover, some studies highlighted the growth of *L. pneumophila* at the level of the outlets of water systems, such as showers, faucets, and toilet flushing [17]. From these points, a *Legionella*-laden aerosol can be produced, causing people exposure by inhalation. Therefore, *Legionella* is a biological hazard also in drinking water systems, as highlighted by the recently released directive on the quality of water intended for human consumption (Directive EU 2020/2184). The new directive recognizes the high health burden of *Legionella* attributable to inhalation during domestic water uses; thus, it indicates *Legionella* monitoring in the context of risk assessment of domestic distribution systems.

To prevent sporadic or epidemic Legionellosis cases, guidelines for water safety plans are available for premise plumbing systems either in tourist accommodation, in hospitals, and other living and working settings. Such guidelines are based on a careful study of the water system to design and implement control measures, including the continuous monitoring of *Legionella* concentrations in water, to evidence the exceeding of the thresholds corresponding to levels of water management actions [18,19]. Such *Legionella* concentration criteria are generally based on professional judgment and could benefit from a risk-based approach to understand health risk corresponding to different indoor fixture use scenarios [20]. The quantitative microbial risk assessment (QMRA) framework is suitable for such a purpose since it allows the calculation of health risk starting from pathogens monitored (or inferred) in the environment [21]. Overall, the attention on *Legionella* proliferation is focused on some priority premises, such as healthcare facilities and spa pools [19], but such bacteria can proliferate in all human-made building water systems, including long-distance transportation (LDT), since they are equipped with plumbing and toilet water tanks. On passenger trains, the water for washbasin and toilets is stored in water tanks (from 200 to 1800 L) located under the car’s roof, and it flows into the toilets by gravity. Tanks are refilled at stations with drinking water supply, but this requirement cannot be fully insured owing to the characteristics of hydraulic systems onboard; therefore, water is not considered suitable for human consumption. Although such water is not ingested, it can be aerosolized during the usage of sinks for handwashing, and it can create an exposure scenario to *Legionella*. LDT is rarely monitored for *Legionella* contamination [22] and, consequently, possible infection or illness risk has not yet been explored. To fill this gap of knowledge, the aims of this paper were: (i) to investigate *Legionella* contamination on LDT through the analysis of 6-year monitoring data from the faucet of toilet cabins and (ii) to develop a QMRA model for sink usage on LDT using monitoring data to understand the probability of *Legionella* infection in such exposure scenario. Although aerosols produced by toilet flushing could also contribute to the probability of infection, such a scenario has not been modeled owing to the lack of monitoring data. Moreover, we estimated the infection risk corresponding to different *Legionella* concentration levels in water commonly used for the adoption of control measures.

## 2. Materials and Methods

### 2.1. Water Sampling and Legionella Analysis

In a period of 6 years, periodical monitoring has been carried out from faucets on Italian passenger LDT, with a total of 398 samples from 2012 to 2018 (except for 2016). Microbial detection of *Legionella* was performed according to the Italian guidelines and as previously described [9]. Briefly, 1 L samples were collected in sterile bottles with sodium thiosulphate (0.01%, *w*/*v*) to neutralize residual chlorine in the toilet water supplies. Each sample was filtered through 0.2 μm pore-diameter polycarbonate membranes, with the membrane then resuspended in 10 mL of the same water sample and vortexed. A 5 mL aliquot of the suspension was heat-treated by incubation at 50 °C for 30 min. Aliquots (100 μL) of both the heat-treated and untreated samples were seeded onto glycine vancomycin polymyxin cycloheximide (GVPC)-selective medium and incubated at 36 ± 1 °C for 10 days in a modified atmosphere (2.5% CO_2_). Putative *Legionella* colonies were subcultured on buffered charcoal yeast extract (BCYE) agar and BCYE agar without cysteine. Colonies that grew only in the presence of cysteine were identified as *Legionella*. Then, these colonies were serotyped using a latex agglutination test (Biolife Italiana Srl, Milan, Italy) to identify *L. pneumophila* sg1, *L. pneumophila* sg2–14, and other species of *Legionella* spp. Water samples containing <100 CFU per liter (CFU/L) were considered negative for *Legionella*.

### 2.2. Statistical Analysis

*Legionella* data were log-transformed before performing statistical analysis to calculate geometric mean and standard deviation. *Legionella* contamination, separately for each serogroup and total, was also described in terms of the interquartile range (IQR), considering the first and third quartiles of concentration data. To analyze the annual differences in *Legionella* contamination, a one-way analysis of variance (ANOVA) was performed (results were considered statistically significant when *p* < 0.05). *Legionella* serogroup loads were categorized into four classes according to Italian water quality guidance for *Legionella* spp. control and prevention in the plumbing system for potable water distribution: ≤100 CFU/L, 101–1000 CFU/L, 1001–10,000 CFU/L, and ≥10,001 CFU/L [23]. All the figures were generated with the R program [24]. Boxplot graphs with individual monitoring data were created using the *ggplot2* package. In each graph, the whiskers represent the minimum and maximum values and the boxes from the 1° and 3° quartiles of the dataset. We performed the fitting distributions to the monitoring data of *L. pneumophila* sg1 (hereafter empirical data). The best-fitting probability distribution was selected from three theoretical distributions (Lognormal, Weibull, Gamma), which are commonly used to approximate microbiological data [25]. The fit of theoretical distributions to empirical data was tested by maximum likelihood estimation (MLE), and the quality of the fit was assessed using the Akaike (AIC) criterion. The analysis was performed in the R program with the *fitdistrplus* package [24,25], and the quality of the fit was also shown by goodness-of-fit graphs [26].

### 2.3. QMRA Methodology

QMRA is a formal four-step process that uses the environmental concentration of pathogens and the amount of exposure to an environmental matrix (dose) as inputs and then estimates the associated probability (risk) of an adverse outcome (infection or illness) as an output, using pathogen-specific mathematical functions describing the dose–response relationship [21,27] (Figure 1). Although many different *Legionella* species and strains of *L. pneumophila* can be found in the considered water, for the QMRA model, we chose to focus on *L. pneumophila* sg1, owing to its epidemiological relevance and abundance in environmental samples and the availability of a published dose–response relationship. Moreover, we considered the infection an endpoint for the risk estimate because the probability of illness following infection is strictly related to the host susceptibility, determined by age, sex, smoking habits, other diseases, and so on [28].

#### 2.3.1. Exposure Assessment

The dose was calculated according to the partitioning coefficient approach [29], in which the concentrations of *L. pneumophila* in water were converted into concentrations in the air using an aerosolization ratio (partitioning coefficient (*PC*)) specific for sink exposure. Therefore, the inhaled dose (*D*) was calculated according to Equation (1), and the values used for each parameter of the exposure assessment (*C_water_*, *PC*, *F*_1–8_, *IR*, *ET*) are shown in Table 1.
*D* = *C_water_* * *PC* ^∗^
*F*_1–8_ * *IR* * *ET*(1)
where *D* is the dose of *L. pneumophila* sg1 deposited in the lungs during sink use (number of *L. pneumophila* sg1), *C_water_* is the concentration of *L. pneumophila* sg1 in water collected from the sink (CFU/L), *PC* is the bacterial water to air partitioning coefficient (CFU L/CFU m^3^), *F*_1–8_ is the fraction of aerosols in the respirable diameter (between 1 μm and 8 μm) produced by a sink (%), *IR* is the inhalation rate of air breathed during sink use, and ET is the exposure duration in the toilet cabin during and after sink use (min).

#### 2.3.2. Dose–Response Assessment

The dose–response relationship for *L. pneumophila* among humans has not yet been developed; therefore, data refer to *L. pneumophila* exposure experiments in an animal model, the guinea pigs, which showed alveolar deposition of aerosol particles comparable with humans and with a similar disease course. The probability of infection of *L. pneumophila* was therefore calculated using the following exponential dose–response model (Equation (2)):
*P_inf_* = 1 − e ^−*D* * *r*^(2)
where *P_inf_* is the probability of infection during a single sink use, *D* is the inhaled dose of *L. pneumophila* derived from the exposure assessment, and *r* is the probability of one cell to survive the host barriers and successfully initiate the infection, corresponding to 0.06 for *L. pneumophila* [32].

#### 2.3.3. Risk Characterization and Sensitivity Analysis

The probability of infection owing to inhalation of *L. pneumophila* sg1 was computed using Monte Carlo analysis (Vensim package, Ventana Systems, Inc., Harvard, MA, USA) to capture the variability of input parameters modeled as probability distribution functions: *L. pneumophila* concentration in water, sink partitioning coefficient, and inhalation rate (see Table 1). The Monte Carlo analysis was run for 200 simulations, each one with a random sampling of 10,000 iterations from input parameters of Table 1 varying at random according to their distributions. The final result can be seen as the probability of *L. pneumophila* infection based on 200 independent measures to improve the accuracy of the health risk analysis. The sensitivity analysis was carried out to test the relative importance of the stochastic variables on the model output (infection risk), and it was performed according to the procedure previously described by Federigi et al. (2020) [33]. Briefly, each of the input parameters was varied, once at a time, according to its own probability distribution function (see Table 1) while keeping each of the other input parameters fixed. In particular, parameters modeled as symmetric probability distributions (inhalation rate and sink use duration) were fixed at the average value, while for other distributions (*L. pneumophila* sg1 concentration in water, sink partitioning coefficient), modal values were chosen. Then, five simulations were run: one with all the input parameters held at their fixed values and the other four letting the variation of only one parameter at a time. In this way, six arrays of 10,000 values of the *P_inf_* were obtained. Finally, the relative importance of each input parameter was calculated as the average *P_inf_* value of the pairwise differences (in absolute value) between the simulation with all the input parameters at a constant value and the simulation with that parameter varying.

## 3. Results

### 3.1. Descriptive Evaluation of Monitoring Data

In the study period, *L. pneumophila* was the only species isolated from the water systems of the toilet cabins, with a total of 217 positive samples considering the entire dataset (54.5%, 217/398). Time trend and differences among serogroups are depicted in Figure 2. The annual variability of *L. pneumophila* contamination ranged from 30.4% (14/46) in 2013 to 81.6% (31/36) in 2018. However, when the sampling size increased, approximately half of the samples were contaminated by *L. pneumophila*, namely 50.6% (86/170) and 53.1% (43/81) (Figure 2). Among positive samples, 64% were colonized by *L. pneumophila* sg 1, 27% by *L. pneumophila* serogroups 2–14, while mixed *Legionella* concentration (sg 1 and serogroups 2–14) was obtained in 9% samples. In positive samples, one-way ANOVA analysis indicated that *Legionella* concentration did not differ significantly through the 6-year monitoring period (one-way ANOVA, *p* = 0.32).

On the whole of positive samples (Table 2), the geometric mean of *L. pneumophila* concentrations was 4.39 × 10^3^ ± 4.97 CFU/L. Of the positive samples (100/217), 46% contained an *L. pneumophila* concentration between 1001 and 10,000 CFU/L, and the majority of them were *L. pneumophila* sg 1 (61/100; 61%). However, *L. pneumophila* sg 1 was the most frequently isolated serogroup also in the lower (101–1000 CFU/L) and in the higher concentration categories (≥10,001 CFU/L), with an occurrence of 88.6% (39/44) and 55% (38/69), respectively.

### 3.2. Fitted Distribution for L. pneumophila Serogroup 1

In the perspective of QMRA, statistical analysis was focused on *L. pneumophila* sg 1 (Section 2.3). Since there was no difference in *Legionella* concentration among different years (ANOVA, *p* > 0.05), the entire dataset for *L. pneumophila* sg1 (139 values) was used to derive a theoretical fitted distribution function. Among the tested theoretical distributions, lognormal was the best in fitting the monitoring data considering the AIC as goodness-of-fit criteria, which was the lowest compared to the other distributions. The best estimates of parameters using the MLE method were µ = 8.166848 and σ = 1.521021, which represent the mean and the standard deviation of the associated normal distribution. Overall, the quality of the fit for lognormal distribution can be also appreciated using graphic tools (Figure 3), which gave strong evidence that monitoring data were well approximated by the theoretical distribution with the above-mentioned µ and σ parameters. Namely, the distribution of the empirical data overlaps with the lognormal fitted distribution, both in the probability density function (PDF) and in the cumulative density function (CDF) plots (Figure 3a,b). Likewise, the empirical dataset of the monitoring data did not deviate from the theoretical dataset from the lognormal fitted distribution, when they are plotted against each other in terms of CDF (probability–probability (P-P) plot) or of quantiles (quantile–quantile (Q-Q) plot) (Figure 3c,d).

### 3.3. QMRA Simulation Results

#### 3.3.1. Infection Risk from Inhalation of *L. pneumophila* sg1

The infection risk was calculated from *L. pneumophila* sg1 concentration in the water using monitoring data collected from toilet faucet and modeled as a lognormal curve with parameters estimated by best-fitting analysis (see Section 3.2), while the other input parameters were drawn from distributions or estimated as point values, based on literature, as reported in Table 1. Based on the modeled distributions, the median value of aerosolized *L. pneumophila* sg1 was 5.40 × 10^−3^ CFU/m^3^ (IQR = 4.31 × 10^−4^–6.32 × 10^−2^), of inhalation rate was 0.015 m^3^/min (IQR = 0.014–0.016), and of exposure time was 1 min (IQR = 0.7–1.3). In such exposure scenario, the single use of toilet faucet was responsible for an inhalation dose of *L. pneumophila* sg1 of 3.86 × 10^−5^ CFU (median value), which corresponded to an infection risk of approximately 2 infections/10^6^ exposures, varying in an interquartile range of 2 infections/10^7^ exposures and 3 infections/10^5^ exposures. Since current guidelines for *Legionella* prevention are based on bacterial concentrations in water [19,23], we considered the relationship between infection risk and *L. pneumophila* sg1 load in water collected from the faucets, then we calculated the probability of infection for each concentration category (Figure 4). When *L. pneumophila* sg1 concentration was ≤100 CFU/L, median infection risk was approx. 1 infection/10^6^ exposures (IQR = 1.83 × 10^−7^–1.97 × 10^−6^), increasing to 11 and 70 infections per 10^6^ exposures when *Legionella* load was 101–1.000 CFU/L and 1001–10,000 CFU/L, respectively. Regarding the highest *Legionella* threshold for water management actions (≥10,001 CFU/L), the median infection risk was approx. 5 infections/10^4^ exposures (IQR = 2.23 × 10^−4^–2.30 × 10^−2^).

#### 3.3.2. Sensitivity Analysis Results

For the sensitivity analysis, the fixed values for each input were calculated on an array of 10,000 iterations of their own distribution, as follows: the modal value of *L. pneumophila* sg1 concentration in water was set at 1.8 × 10^4^ CFU/L, the mean value of the inhalation rate at 0.015 m^3^/min, the mean value of sink use duration at 1 min, and the sink partitioning coefficient was set at the modal value of 1.02 × 10^−5^ L/m^3^. The most impacting parameter was *L. pneumophila* sg1 concentration followed, in order, by sink partitioning, sink use duration, and inhalation rate (Figure 5). Such results could be attributable to the different ranges of variation of input parameters; for instance, the range of variation of sink use duration (two times its minimum value) was wider than that of the inhalation rate (about 1/3 of the minimum value).

## 4. Discussion

Legionellae are naturally occurring bacteria in any aquatic environment, whose pathogenic potential is mainly attributed to *L. pneumophila* [34]. From a public health perspective, particular attention is dedicated to *L. pneumophila* sg 1 since it is the most prevalent reported strain of *Legionella* in building water systems [35] that is currently the predominant serogroup in clinical isolates, accounting for approximately 85% of cases confirmed by culture worldwide [36,37,38]. Plumbing systems of LDT can be a source of *Legionella*, but there is a paucity of environmental surveillance data. At the time of our study, we found only one paper carried out on passenger trains, collecting water from plumbing and toilet water tanks before and after chlorine-based disinfection treatment [22]. Our monitoring results showed similar percentages of *Legionella* positivity, species, and serogroups as in pre-decontamination samples of Quaranta et al. 2012 [22]; namely, they found more than 50% of water samples positive for *Legionella*, they identified only the *L. pneumophila* species, and the most frequently isolated was *L. pneumophila* sg1. Regarding the *Legionella* concentrations, our study found slightly higher values, but still with a geometric means around 10^3^ CFU/L.

To date, health risk coming from sink exposure is rarely modeled since studies have focused on other residential water uses, mainly having a shower, bathtub, and whirlpool [39,40,41,42]. In this paper, we focused on sink exposure since such a scenario could be relevant for the LDT, and monitoring data were available to avoid adapting data collected from the literature. Available *L. pneumophila* load referred to concentration in water; thus, the dose was calculated using a sink-specific aerosolization ratio because it can vary based on the type of water fixture (i.e., shower, sink, toilet, pool spa, whirlpool), as thoroughly described by Hines et al. (2014) [43]. Although the water provided on LDT is not considered potable, the risk assessment and management for *Legionella* can be carried out as in the case of drinking water distribution systems (Directive EU 2020/2184), owing to the similarities of the water plant configuration and of critical points for the proliferation. Currently, management actions for *Legionella* prevention in community or healthcare settings are based on *Legionella* spp. detection in water collected from the fixtures or collection tanks of the plumbing system. The thresholds for the adoption of control measures usually range from 10^2^ to 10^5^ CFU per liter [18,19], with the above-mentioned Directive EU 2020/2184 setting a parametric value of 1000 CFU/L for applying restriction of the use and remedial actions to drinking water distribution systems. However, such values are not established on a risk-based criterion [20], and specific guidelines to prevent *Legionella*’s transmission on LDT are currently missing. Therefore, we performed a quantitative assessment to understand the infection risk associated with *Legionella* thresholds in water on LDT, using Italian guidelines as an example for deriving concentration categories. The model outcomes showed a very low risk when *L. pneumophila* sg1 concentration was ≤100 CFU/L, with approximately 1 infection per 1 million exposed passengers. However, monitoring results revealed that around 30% of samples were ≥10,000 CFU/L, and, in that case, infection risk increased up to 5 infections per 10,000 exposed (median value). Although such health risk is overall low, the obtained results deserve consideration since public transport is commonly used by populations with risk factors for the development of a severe form of the disease, such as older age, surgery, immunodeficiency, or smoking habit. Moreover, personnel employed on LDT (i.e., train drivers, cleaning staff onboard) could have a prolonged exposure than passengers. This topic is currently little explored, but a seroprevalence study on Turkish bus drivers revealed high *L. pneumophila* antibody levels among such workers, demonstrating a chronic exposure to the pathogen, probably coming from bus air-conditioning systems [44].

### Limitations of the Study

In this paper, the *Legionella* risk assessment on LDT was carried out considering the faucets as the only source of *Legionella*-laden aerosol. Nevertheless, additional exposure could derive also from toilet flushing, which was not modeled owing to the lack of monitoring data. Another limitation of the study could be that illness risk was not calculated, and health risk was expressed as the probability of infection. However, the development of illness can greatly vary depending on the underlying health of individuals, with little understanding of the mechanisms responsible for the development of self-limiting symptoms (Pontiac fever) or the severe pneumonia outcome (Legionnaire’s disease) [28]. Regarding the trend of *Legionella* contamination over time, we considered together *L. pneumophila* sg1 values coming from various annual monitoring campaigns for statistical fitting of data. Nevertheless, we found an annual variability in serogroups occurrence that was not further addressed since it goes beyond the aim of the present study. Therefore, the analysis of time trend of *L. pneumophila* contamination could be further investigated considering not only sampling size for each investigated period but also variables, both technical (e.g., renewal of trains, disinfection systems) and environmental (e.g., climate) ones.

## 5. Conclusions

This paper addressed a currently little-explored topic represented by the risk assessment of *Legionella* on LDT, using monitoring data collected from sinks of toilet cabins. Monitoring results on faucets showed that water systems on LDT can be colonized by *Legionella*, which was responsible for a low number of infections acquired through the studied exposure scenario. Nevertheless, some samples showed a very high bacterial load, which in turn was associated with an increase in the infection risk. Overall, this paper highlighted the importance of *Legionella* risk assessment also on LDT, which could be useful for planning adequate control measures to protect human health, both of passengers and of workers.

## Figures and Tables

**Figure 1 ijerph-19-01960-f001:**
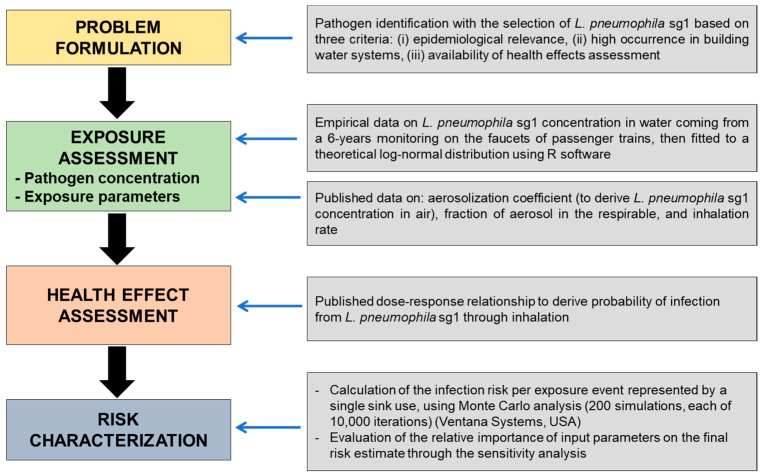
Schematics of the QMRA model framework and model input parameters for the modeled exposure scenario of sink usage on LDT.

**Figure 2 ijerph-19-01960-f002:**
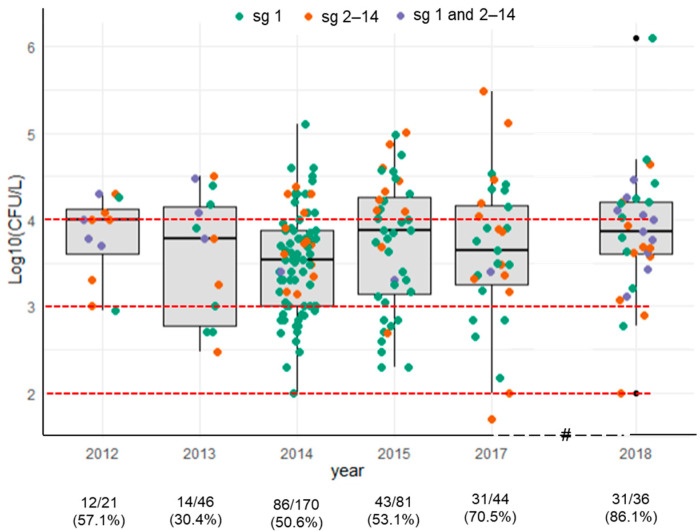
Annual variability of *L. pneumophila* concentrations in positive samples. For each year, the distribution of the whole *Legionella* concentration values is represented by boxplot and the serogroups are identified by different colors. Red dashed lines correspond to *Legionella* thresholds in water for the application of control measures in healthcare and community settings, according to Italian guidelines [23].

**Figure 3 ijerph-19-01960-f003:**
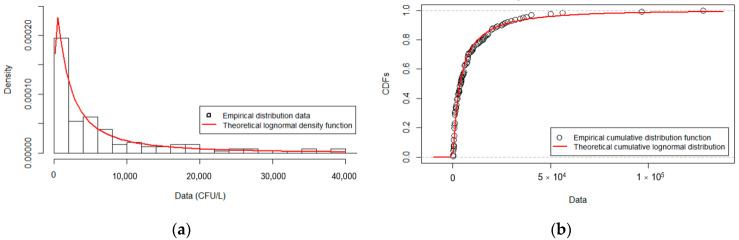
Results of the fitting of a Lognormal distribution to the monitoring dataset. Empirical and theoretical distributions are represented as probability density function (PDF) (**a**) or as cumulative density function (CDF) (**b**). Empirical data set plotted against the theoretical lognormal data set are represented as P-P plot (**c**) and Q-Q plot (**d**). A straight blue line represents perfect fitting in (**c**,**d**).

**Figure 4 ijerph-19-01960-f004:**
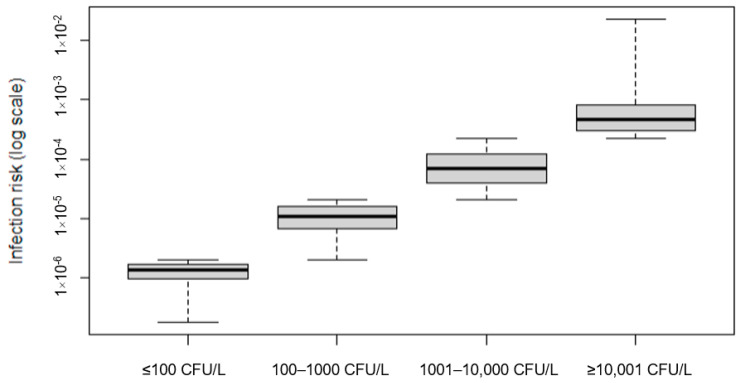
Estimated probability of infection for single use of faucet for different concentrations of *L. pneumophila* sg1 in water, according to concentration categories for water management actions.

**Figure 5 ijerph-19-01960-f005:**
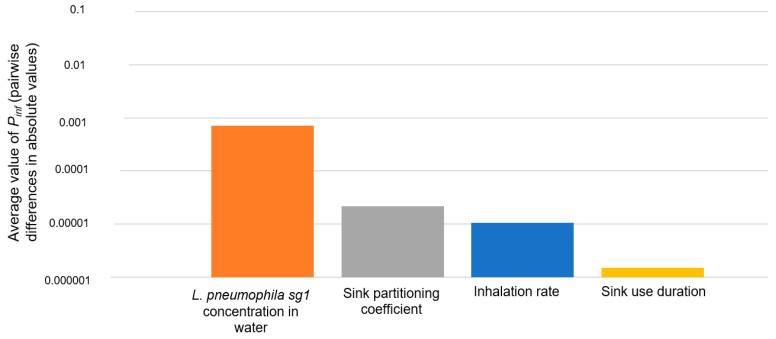
Sensitivity analysis results for the QMRA model. The histograms represent the average values of the pairwise differences of *P_inf_* when all the input parameters are fixed and one varying at a time.

**Table 1 ijerph-19-01960-t001:** Distributions and parameters of the exposure assessment in the QMRA framework.

Input Variables	Description	Unit	Characterization	Source and Comments
*C_water_*	*L. pneumophila* sg1 concentration in water	CFU/L	lognormal distribution ^1^ (µ = 8.166848, σ = 1.521021)	This article, based on a 6-year monitoring period
*PC*	Sink partitioning coefficient	L/m^3^	lognormal distribution ^1^ (μ = −13.3, σ = 3.49) truncated on the interval [0, 2.35 × 10^−3^]	Hamilton et al., 2019 [20]. Data analysis of 19 paired water and air samples from hot-water faucets
*F* _1–8_	Percentage of aerosols in respirable range (between 1 and 8 μm reported) for partitioning coefficient	%	point estimate (50)	Bollin et al., 1985 [30]. Monitoring of air samples from hot-water faucets, and approximately half of recovered *Legionella* were between 1 and 8 μm aerosol
*IR*	Inhalation rate	m^3^/min	uniform distribution (min = 0.013, max = 0.017)	USEPA 2011 [31]. Inhalation rate for individuals engaging in light activities
*ET*	Sink use duration	min	uniform distribution (min = 0.5, max = 1.5)	An assumption on the duration of an individual would stay in the toilet for hand washing.

^1^ The lognormal distribution of each variable Y has been evaluated as exp(µ + σ*Z) where Z is a standardized normal variable with mean 0 and standard deviation 1, and µ and σ are, respectively, the mean and the standard deviation of a generic normal distribution.

**Table 2 ijerph-19-01960-t002:** Contamination of *L. pneumophila* in positive samples. For load distribution categories, percentages refer to the column (calculation based on the type of serogroup).

	*L. pneumophila* Total	*L. pneumophila* sg1	*L. pneumophila* sg 2–14	Mixed *L. pneumophila* sg 1 and sg 2–14
Positive samples (n°, %)	217/398 (54.5%)	139/217 (64.1%)	58/217 (26.7%)	20/217 (9.2%)
**Count (CFU/L)**				
Geometric mean	4.93 × 10^3^ ± 4.97	3.67 × 10^3^ ± 4.96	5.72 × 10^3^ ± 5.74	7.00 × 10^3^ ± 2.45
Median	5.00 × 10^3^	4.10 × 10^3^	6.68 × 10^3^	6.63 × 10^3^
IQR (1°–3° quartiles)	1.50 × 10^3^–1.30 × 10^4^	1.00 × 10^3^–1.14 × 10^4^	2.13 × 10^3^–1.93 × 10^4^	3.79 × 10^3^–1.22 × 10^4^
**Load distribution categories (n°, %)**				
≤100 CFU/L	4/217 (1.8%)	1/139 (0.7%)	3/58 (5.2%)	0/20 (0%)
101–1000 CFU/L	44/217 (20.3%)	39/139 (28.1%)	5/58 (8.6%)	0/20 (0%)
1001–10,000 CFU/L	100/217 (46.1%)	61/139 (43.9%)	27/58 (46.6%)	12/20 (60%)
≥10,001 CFU/L	69/217 (31.8%)	38/139 (27.3%)	23/58 (39.7%)	8/20 (40%)

## Data Availability

Data available on request due to privacy restrictions.

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
