# Peer review of "Quantitative Microbial Risk Assessment Applied to Legionella Contamination on Long-Distance Public Transport"

_ijerph, 2022, doi:10.3390/ijerph19041960_

Round 1

Reviewer 1 Report

The study was well planned and the results were well interpreted. Minor technical corrections are needed.

  • Many researchers scan articles by reading only the abstracts. In manuscript abstract, the term "that should be consider for strongly reducing the risk for travelers and workers health on LDT." should be made ambiguous in the abstract. Although the risk of legionellosis has been found to be low on study, the risk also depends on the immunity status of the host. The abstract should be revised at this step.
  • Although the "R language" text can be accepted as this work, add-on packages on the R program are used. Therefore, it is sufficient to say "R program" instead of "R language".
  • In "Figure 1", please include the references to the cited in the colored boxes. It should also be added to "references" section if not exist.
  • Please adding, number of samples (N) and positivity rate (%) in Figure 2, at the x-title position with each year. 
    please, specify dashed axis on x-axis for year 2016, like as  "-------#------".

  • Please adding serological testing manufacturer.
  • Only Legionella sg1 (not all) isolates have been found at the highest rate. However, it is low in 2012 and 2013. Could this have influenced the results of the study? The increase in Legionella sg1 isolation over time does not seem to be technique dependent. What could be the reason for the high isolation rate in 2014? Change over the years is not the main objective of the study. However, this situation seen in Figure 1 should be discussed in the discussion and/or limitation section. 
    Please adding in the manuscript whether there is an environmental difference (renewal of trains, a situation that will affect the operation during maintenance and repair) during the sampling period.
  • The same analysis was not performed for Legionella serogroups other than sg1. Although sg1 is known to be more virulent, it should be added why only sg1 was included in the statistical analysis (or only isolates other than sg1 were excluded) in the study.

best regards,

Author Response

Point 1: Many researchers scan articles by reading only the abstracts. In manuscript abstract, the term "that should be consider for strongly reducing the risk for travelers and workers health on LDT." should be made ambiguous in the abstract. Although the risk of legionellosis has been found to be low on study, the risk also depends on the immunity status of the host. The abstract should be revised at this step.

Response to point 1: We thank the Reviewer for this suggestion. We modified the text as follows:Our QMRA results showed a low Legionella infection risk from faucets on LDT, however it deserves consideration since LDT can be used by people highly susceptible for the development of severe form of the disease, owing to their immunological status or other predisposing factors(page 1, lines 29-32).

Point 2: Although the "R language" text can be accepted as this work, add-on packages on the R program are used. Therefore, it is sufficient to say "R program" instead of "R language".

Response to point 2: Done, thanks for the suggestion (page 3, lines 119 and 128 of the revised manuscript).

Point 3: In "Figure 1", please include the references to the cited in the colored boxes. It should also be added to "references" section if not exist.

Response to point 3: We thank the Reviewer for pointing out such typos in Figure 1. All references were already included in the reference list and the figure has been modified accordingly, using numbering (page 4, line 144 of the revised manuscript).

Point 4: Please adding, number of samples (N) and positivity rate (%) in Figure 2, at the x-title position with each year. 
please, specify dashed axis on x-axis for year 2016, like as  "-------#------".

Response to point 4: Figure 2 has been modified accordingly (page 6, line 216 of the revised manuscript) by the addition of number of samples and positivity rate below each year of the x-axis.

Point 5: Please adding serological testing manufacturer.

Response to point 5: We thank the Reviewer for the comment. We added the manufacturer as follows: “Then, these colonies were serotyped using a latex agglutination test (Biolife Italiana Srl, Milan, Italy) to identify L. pneumophila sg1, L. pneumophila sg2-14, other species of Legionella spp.(page 3, lines 105-106 of the revised manuscript).

Point 6: Only Legionella sg1 (not all) isolates have been found at the highest rate. However, it is low in 2012 and 2013. Could this have influenced the results of the study? The increase in Legionella sg1 isolation over time does not seem to be technique dependent. What could be the reason for the high isolation rate in 2014? Change over the years is not the main objective of the study. However, this situation seen in Figure 1 should be discussed in the discussion and/or limitation section. Please adding in the manuscript whether there is an environmental difference (renewal of trains, a situation that will affect the operation during maintenance and repair) during the sampling period.

Response to point 6: We thank the Reviewer for the opportunity to clarify this point. The amount of collected samples varied over time, therefore the L. pneumophila detection rate depended on the total amount of samples collected in each year. To clarify this point, we performed a series of changes, as listed below:

  1. Modification of Figure 2, by adding detection rate of L. pneumophila (below each year on x-axis of the figure), as suggested by the Reviewer in the comment above;
  2. New text in the Result (Section 3.1):In the study period, L. pneumophila was the only species isolated from the water systems of the toilet cabins, with a total of 217 positive samples considering the entire dataset (54.5%, 217/398). Time-trend and differences among serogroups are depicted in Figure 2. The annual variability of L. pneumophila contamination ranged from 30.4% (14/46) in 2013 to 81.6% (31/36) in 2018. However, when the sampling size increased, approximately half of the samples were contaminated by L. pneumophila, namely 50.6% (86/170) and 53.1% (43/81) (Figure 2).(page 6, lines 203-209);
  3. New text in the Discussion (Section 4.1):Regarding the trend of Legionella contamination over time, we considered together L. pneumophila sg1 values coming from various annual monitoring campaigns for statis-tical fitting of data. Nevertheless, we found an annual variability in serogroups occur-rence, that was not further addressed since it goes beyond the aim of the present study. Therefore, the analysis of time-trend of L. pneumophila contamination could be further investigated considering not only sampling size for each investigated period, but also variables both technical (e.g., renewal of trains, disinfection systems) and environ-mental (e.g., climate) ones.(page 10, lines 357-365 of the revised manuscript).

Point 7: The same analysis was not performed for Legionellaserogroups other than sg1. Although sg1 is known to be more virulent, it should be added why only sg1 was included in the statistical analysis (or only isolates other than sg1 were excluded) in the study.

Response to point 7: We thank the Reviewer for the opportunity to clarify the reason of the choice of L. pneumophila sg1. Thus, at the beginning of the Section 3.2, we added the following sentence:In the perspective of QMRA, statistical analysis was focused on L. pneumophila sg 1 (Section 2.3).” (page 7, lines 231-232 of the revised manuscript).

Reviewer 2 Report

The paper is good and well written.

Author Response

We thank the Reviewer for this the positive feedback, we did our best to develop a good quality and original manuscript.

Reviewer 3 Report

Materials and methods. The study is well structured and methodologically consistent. However, in my opinion, the statistics part contains too many details; I would therefore suggest a more concise presentation of mathematical modeling.

Results. What would be the cause or causes that influence the annual variability of water contamination in water tanks? More explanations could lead to measures to limit Legionella contamination. I also think that more explanations in connection with Tables 3 and 4 would be welcome.

Conclusions. The authors should emphasize more strongly the original results obtained and their practical applicability. With the changes suggested above, the article may be published in International Journal of Environmental Research and Public Health.

Author Response

Please see the attachment (some changes are reported as Tables to compare the previous versione with the revised ones).
